# A Density Functional Theory and Microkinetic Study of Acetylene Partial Oxidation on the Perfect and Defective Cu_2_O (111) Surface Models

**DOI:** 10.3390/molecules27196748

**Published:** 2022-10-10

**Authors:** Ling-Nan Wu, Zhen-Yu Tian, Wu Qin

**Affiliations:** 1Institute of Engineering Thermophysics, Chinese Academy of Sciences, Beijing 100190, China; 2University of Chinese Academy of Sciences, Beijing 100049, China; 3Engineering Laboratory for Biomass Generation Equipment, North China Electric Power University, Beijing 102206, China

**Keywords:** acetylene, partial oxidation, density functional theory calculations, Cu_2_O (111) surface, defects

## Abstract

The catalytic removal of C_2_H_2_ by Cu_2_O was studied by investigating the adsorption and partial oxidation mechanism of C_2_H_2_ on both perfect (stoichiometric) and Cu_CUS_-defective Cu_2_O (111) surface models using density functional theory calculations. The chemisorption of C_2_H_2_ on perfect and defective surface models needs to overcome the energy barrier of 0.70 and 0.81 eV at 0 K. The direct decomposition of C_2_H_2_ on both surface models is energy demanding with the energy barrier of 1.92 and 1.62 eV for the perfect and defective surface models, respectively. The H-abstractions of the chemisorbed C_2_H_2_ by a series of radicals including H, OH, HO_2_, CH_3_, O, and O_2_ following the Langmuir–Hinshelwood mechanism have been compared. On the perfect Cu_2_O (111) surface model, the activity order of the adsorbed radicals toward H-abstraction of C_2_H_2_ is: OH > O_2_ > HO_2_ > O > CH_3_ > H, while on the defective Cu_2_O (111) surface model, the activity follows the sequence: O > OH > O_2_ > HO_2_ > H > CH_3_. The Cu_CUS_ defect could remarkably facilitate the H-abstraction of C_2_H_2_ by O_2_. The partial oxidation of C_2_H_2_ on the Cu_2_O (111) surface model tends to proceed with the chemisorption process and the following H-abstraction process rather than the direct decomposition process. The reaction of C_2_H_2_ H-abstraction by O_2_ dictates the C_2_H_2_ overall reaction rate on the perfect Cu_2_O (111) surface model and the chemisorption of C_2_H_2_ is the rate-determining step on the defective Cu_2_O (111) surface model. The results of this work could benefit the understanding of the C_2_H_2_ reaction on the Cu_2_O (111) surface and future heterogeneous modeling.

## 1. Introduction

Acetylene (C_2_H_2_) is a kind of important industrial raw materials used for various purposes, including oxyacetylene welding, cutting, illuminant, soldering metals, signaling, precipitating metals, particularly copper, manufacture of acetaldehyde, acetic acid, etc. [1]. C_2_H_2_ is a significant intermediate formed during the combustion of hydrocarbons, especially under fuel-rich conditions, which is responsible for the soot formation during combustion processes via the H-abstraction-C_2_H_2_-addition (HACA) mechanism [2,3]. The production of C_2_H_2_ during combustion could endanger the safety and efficiency of combustors, etc. C_2_H_2_ is also a component of volatile organic compounds (VOCs), and it gains increasing attention due to its toxicity to the environment and human health. Exposure to high concentrations of C_2_H_2_ may cause loss of consciousness or even death, and it is a serious fire and explosion hazard. C_2_H_2_ is an undesirable by-product of the petroleum cracking process, and it causes damage to the catalyst for the ethylene polymerization process [4]. To address the problems caused by C_2_H_2_ formation and emission, the efficient removal of C_2_H_2_ during combustion and industrial processes is of great interest. To the best of our knowledge, much attention has been paid to the study of C_2_H_2_/hydrocarbons homogeneous kinetics under pyrolysis [5,6], oxidation [3,7], and flame conditions [2], and kinetic models predicting the reaction characteristics under these conditions have been proposed [3,7], while relatively less attention has been paid to the heterogeneous processes of C_2_H_2_.

Catalytic removal is an important technique in exhaust gas purification, and the activity of catalysts plays a crucial role in the catalytic process. Previous studies have shown that Cu_2_O thin film catalysts prepared by the pulsed-spray evaporation chemical vapor deposition (PSE-CVD) method are effective for the catalytic removal of C_2_H_2_ [8], but the oxidation mechanism of C_2_H_2_ on the Cu_2_O surface remains unclear. Theoretical studies based on density functional theory (DFT) calculations have been widely used as an effective tool in revealing the gas-surface heterogeneous reaction mechanisms on Cu-based oxide surfaces [9,10,11,12,13,14,15,16,17,18,19,20,21,22,23,24,25,26]. DFT studies regarding the C_2_H_2_ hydrogenation process have been reported previously. Zhang et al. [27,28] have studied C_2_H_2_ hydrogenation to ethylene using DFT calculations, and it is found that the valence state of the surface Cu site has an important impact on the surface catalytic ability toward the C_2_H_2_ hydrogenation to ethylene. Good command of the surface oxidation mechanism could be beneficial for the development of high-performance catalysts. Experiments could provide useful information by studying the dependency of catalysts’ performance on the preparation methods and macroscopic parameters, such as temperature, pressure, PH, etc. Proper characterizing techniques, such as SEM, XPS, XRD, etc., could also throw light upon the surface morphology and surface properties, which could help better understand the nature of the catalytic process. In addition, theoretical studies based on DFT calculations could also reveal the intrinsic surface reaction mechanism, and the effect of surface sites and vacancies on the catalytic performance is still needed. The establishment of a proper surface model is of importance for the theoretical investigation of the C_2_H_2_ partial oxidation process on the Cu_2_O surface. The Cu_2_O (111) plane is the most widely used surface model to reveal the heterogeneous reaction mechanism on the Cu_2_O surface due to its thermodynamic stability, while many of them used the bulk-terminated models (the stoichiometric surface model or the perfect surface model) [13,16,29,30,31] but not the more stable Cu_2_O (111)–Cu_CUS_ surface model as proposed by Soon et al. [32], and the importance of the Cu_CUS_ vacancy has also been confirmed by Önsten et al. [33] experimentally. Our previous study also found that the defective Cu_2_O (111)–Cu_CUS_ surface model could improve the surface activity toward CO oxidation [34] than the perfect one. Therefore, it is of significance to consider the surface defects when studying Cu_2_O surface chemistry.

To provide a better understanding of the reaction mechanism of C_2_H_2_ on the Cu_2_O surface, the adsorption and reaction processes of C_2_H_2_ on the Cu_2_O surface models were studied based on DFT calculations in this study. The reaction processes of C_2_H_2_ on the Cu_2_O surface models, including the adsorption process, decomposition process, and H-abstraction reactions, by a variety of radicals and O_2_ have been studied. The effect of the surface defect on the C_2_H_2_ elementary reaction steps has been explored by studying the C_2_H_2_ conversion on both the stoichiometric perfect Cu_2_O (111) surface model and the Cu_2_O (111)-Cu_CUS_ defective surface model. The rate constants have been calculated and the parameters are provided in the Arrhenius form, which could be helpful for heterogeneous kinetic modeling studies.

## 2. Computational Details

DFT calculations were performed using the DMol^3^ code [35,36]. The generalized gradient approximation (GGA) functional of Perdew–Burke–Ernzerhof (PBE) [37] was used for exchange and correlation potential. The double numerical basis set plus the polarization (DNP) basis set was used for all the calculations. The DFT semi-core pseudopots (DSPP) core treatment was used for inner core pseudopotential treatment, which introduces relativistic correction into the cores. Transition state structures were preliminarily searched by the combination of linear synchronous transit (LST) and quadratic synchronous transit (QST) method and then optimized using the eigenvector following (EF) method to validate only one imaginary vibrational mode, which corresponds to a first-order saddle point on the potential energy surface and correctly connects the reactant and the product of each elementary reaction.

A higher computational accuracy has been used in the current work compared with our previous works [29,34]. The orbital cut-off has increased from 4.0 to 4.4 Å, and the convergence threshold of the self-consistent field (SCF) has increased to 1.0 × 10^−6^ from 1.0 × 10^−5^. The convergence criteria of energy, maximum force, and maximum displacement are 1.0 × 10^−5^ Ha, 0.002 Ha/Å, and 0.005 Å, respectively. For the crystal optimization, a 4 × 4 × 4 Monkhorst–Pack k-point grid was used. The surface planes were built by cleaving the Cu_2_O (111) surface from the optimized Cu_2_O crystal. A sheet of 10 Å vacuum layer was placed over the surface slab to avoid interference from imaging surface planes due to the periodic boundary conditions. The defective surface model was established by removing the top and bottom layer unsaturated Cu_CUS_ sites. A 3 × 3 × 1 Monkhorst–Pack k-point grid was used for the energy calculations of the succeeding surface reactions.

Adsorption energy ($E_{\mathrm{ad}}$) is used to evaluate the interaction between the surface and the adsorbate, which is defined as:(1)$$E_{\mathrm{ad}}=E_{\mathrm{sys}}-E_{\mathrm{ads}}-E_{\mathrm{sur}}$$
where $E_{\mathrm{sys}}$ is the energy of the system after adsorption; $E_{\mathrm{ads}}$ is the energy of the adsorbate before adsorption; $E_{\mathrm{sur}}$ is the energy of the clean surface before adsorption.

The Gibbs free energy of activation was calculated by combining zero-point energy (ZPE), the electronic energies calculated at 0 K, and the thermal corrections at elevated temperatures. For surface species, the translations and rotations were converted into frustrated oscillation modes and were included in the vibration analysis [38]. Reaction rate constants of elementary reaction steps were calculated based on harmonic transition-state theory (HTST) [29,34,38,39], which is $k=\frac{k_{B}T}{h}\left(\frac{-\Delta G_{a}}{RT}\right)$, where *k* is reaction rate constant, *k*_B_ is the Boltzmann constant, *T* is temperature, *h* is the Planck constant, *R* is the universal gas constant, and $\Delta G_{a}$ is the Gibbs free energy of activation. Detailed calculation processes can be found elsewhere [34,40].

## 3. Results and Discussion

### 3.1. Perfect and Defective Cu_2_O (111) Surface Models and C_2_H_2_ Adsorption

The (111) surface plane of Cu_2_O crystal has been used throughout this study as it is the most thermodynamically stable and, therefore, dominantly exposed low-index surface plane [29,34,41,42]. The perfect and the Cu_CUS_-defective Cu_2_O (111) surface models have been established, as shown in Figure 1. The lattice constant of the perfect Cu_2_O (111) crystal after geometric optimization is 4.33 Å as a result of the improved convergence accuracy, which is close to the previously reported values (4.32 Å [43]) and experimental values (4.27 Å [23]). The perfect Cu_2_O (111) surface model is stoichiometric with the Cu/O ratio to be exactly two, and it contains four kinds of surface top-sites on the top layer, including the saturated copper (Cu_CSS_) site, the saturated oxygen (O_CSS_) site, the unsaturated copper (Cu_CUS_), and the unsaturated oxygen (O_CUS_) site, while the defective Cu_2_O (111) surface model only comprises the Cu_CSS_, O_CSS_, and O_CUS_ sites, as the top and bottom Cu_CUS_ sites are missing. The Cu_CUS_ sites are active in absorbing the molecules, but the strong covalent bond between the adsorbate and the Cu_CUS_ sites may hinder the following surface reactions.

The stable adsorption structure of C_2_H_2_ on the Cu_2_O (111) surface was explored by comparing the adsorption energies of C_2_H_2_ on different surface sites of the Cu_2_O (111) surface models. One C_2_H_2_ molecule was placed on different surface sites, including the surface Cu_CUS_ site, the Cu_CSS_ site, the O_CUS_ site, and the O_CSS_ site, and the adsorption energies were obtained after the geometric optimization process. Only the most stable adsorption structure corresponding to the largest adsorption energy is presented.

The adsorption processes of C_2_H_2_ on the perfect and defective Cu_2_O (111) surface models are shown in Figure 2. For the adsorption on the perfect Cu_2_O (111) surface model, a C_2_H_2_ molecule will first adsorb over the surface unsaturated Cu_CUS_ site with the energy release of 1.13 eV, and the linear structure of C_2_H_2_ is slightly distorted with the O–C–C angles decreasing from both 180° to 162° and 169°. The C–C bond length of C_2_H_2_ increases to 1.242 from 1.211 Å in the gas phase and the C-H bond length also increases to 1.077 and 1.081 from 1.071 Å. The activated C_2_H_2_ molecule will then overcome the energy barrier of 0.70 eV and interacts with one surface lattice O_CUS_ site and its neighboring Cu_CUS_ site and one Cu_CSS_ site, forming a chemisorbed structure depicted as FS in Figure 2a with the heat release of 1.78 eV in total. For the same process on the defective Cu_2_O (111) surface model. One C_2_H_2_ molecule will first undergo a physisorption process releasing 0.20 eV. The bond length of the C–H bond close to the surface increases to 1.076 Å, while the bond lengths of the other C–H bond and the C–C bond are almost unchanged after physisorption. The physisorbed C_2_H_2_ will then overcome the energy barrier of 0.81 eV and react with the surface O_CUS_ site to form an adsorbed CHCHO* species, which is bonded to three surface Cu_CSS_ sites. The whole reaction process on the defective Cu_2_O (111) surface model releases 1.26 eV. By comparison, the perfect Cu_2_O (111) surface model is more favorable for the C_2_H_2_ adsorption at 0 K with a lower energy barrier and larger energy release, which is due to the existence of the active Cu_CUS_ site in activating the C_2_H_2_ bond.

### 3.2. Decomposition of C_2_H_2_ on the Perfect and the Defective Cu_2_O (111) Surface Models

The direct decomposition processes of the chemisorbed C_2_H_2_* molecule undergoing the cleavage of the C-H bond on the Cu_2_O (111) surface models are first investigated, which represents the surface activity toward C_2_H_2_ direct decomposition when there are no other adsorbates on the surface. Reaction energy profiles and the structures of the initial states, transition states, and final states are provided in Figure 3. The energy barrier of the reaction process on the perfect Cu_2_O (111) surface model is 1.92 eV, and the reaction is an exothermic process releasing 0.21 eV. The adsorbed C_2_H_2_ molecule will undergo an H-abstraction process, and the H atom will shift to the surface Cu_CUS_ site after the H-abstraction process. The C_2_H part will react with the lattice O_CUS_ site and form an adsorbed HCCO* species on the surface after the reaction. The H-C-C part of the formed HCCO* species has a nearly linear structure with the H-C-C angle to be 177°. As for the direct decomposition of C_2_H_2_ on the defective Cu_2_O (111) surface model, the energy barrier has increased to 2.46 eV, and the reaction is an endothermic process adsorbing 0.36 eV. Therefore, the decomposition of the chemisorbed C_2_H_2_ on both the perfect and the defective Cu_2_O (111) surface models is hard to happen in terms of the reaction energy barrier, and the perfect Cu_2_O (111) surface model is more favorable than the defective one comparatively, which is due to the existence of the neighboring active Cu_CUS_ site.

### 3.3. H-Abstraction Reactions of Chemisorbed C_2_H_2_ by O_2_ on the Cu_2_O (111) Surface Models

The H-abstraction reactions are important in the consumption of C_2_H_2_, hence the H-abstraction of the chemisorbed C_2_H_2_ by O_2_ was studied in this section. The Langmuir–Hinshelwood reaction mechanism featuring the reaction between two adsorbed molecules on the perfect and defective Cu_2_O (111) surface models is studied. The left part of Figure 4 shows the H-abstraction process on the perfect Cu_2_O (111) surface model. The surface unsaturated Cu_CUS_ site is active for O_2_ adsorption, and an O_2_ molecule will adsorb on the Cu_CUS_ site close to the chemisorbed C_2_H_2_ molecule, releasing 1.09 eV. Then, one H atom of the chemisorbed C_2_H_2_ will transfer to the adsorbed O_2_ forming an adsorbed HO_2_ molecule by overcoming the energy barrier of 1.43 eV. The reaction releases 0.23 eV with the formation of an adsorbed HO_2_ and an adsorbed HCCO species on the surface.

The energy profile of C_2_H_2_ H-abstraction by O_2_ on the defective Cu_2_O (111) surface model is shown in the right part of Figure 4. An O_2_ molecule will first adsorb on the surface hollow site, and the chemisorbed C_2_H_2_ needs to overcome the energy barrier of 0.97 eV to react with the adsorbed O_2_ and form an HO_2_ on the hollow site with an energy release of 0.86 eV. An adsorbed HCCO species is formed after the reaction on top of three Cu_CSS_ sites. Compared with the perfect surface model, the H-abstraction of C_2_H_2_ by O_2_ on the defective Cu_2_O (111) surface is easier with a lower energy barrier and a larger energy release. The homogeneous H-abstraction of C_2_H_2_ by O_2_ is also calculated for comparison purposes, and the reaction is a strong endothermic process adsorbing 3.59 eV. Therefore, both the perfect and the defective Cu_2_O (111) surface models are favorable for the H-abstraction process of C_2_H_2_ by O_2_ from the thermodynamic point of view, and the Cu_CUS_ defect will be beneficial for the H-abstraction of C_2_H_2_ by O_2_.

### 3.4. H-Abstraction and H-Addition of Chemisorbed C_2_H_2_ by Atomic H

The reaction between a chemisorbed C_2_H_2_ and an adsorbed atomic H via the LH mechanism is further studied in this section. The neighboring adsorbed atomic H, as shown in Figure 5, is bonded to the surface unsaturated Cu_CUS_ site, and it could attack the chemisorbed C_2_H_2_ and form an adsorbed H_2_ and an adsorbed HCCO species together with a lattice O_CUS_. The reaction process is endothermic with energy adsorption of 0.78 eV, and the energy barrier is 2.75 eV. The chemisorbed C_2_H_2_ could also react with an adsorbed H together with the lattice O to form a CH_2_CHO species, as shown in the right part of Figure 5, on the perfect Cu_2_O (111) surface model. The energy barrier of the reaction process is 1.78 eV, which is lower than that of the H-abstraction reaction of the adsorbed C_2_H_2_ with an energy barrier of 2.75 eV. In terms of the reaction energy, the formation of CH_2_CHO needs to adsorb 0.67 eV, while the H-abstraction process adsorbs 0.78 eV. Therefore, the adsorbed C_2_H_2_ is more likely to be converted to CH_2_CHO when reacting with an adjacent adsorbed H together with a lattice O_CSS_ site on the perfect Cu_2_O (111) surface model.

### 3.5. A Comparison of H-Abstraction Reactions of Chemisorbed C_2_H_2_ by Different Radicals

The energy profile of the interaction between the chemisorbed C_2_H_2_ species and the pre-adsorbed oxygen molecule and radicals, including H, OH, O, HO_2_, and CH_3_, on the perfect Cu_2_O (111) surface model are compared in Figure 6. The IS state also incorporates the energy release of various radicals on the surface unsaturated Cu_CUS_ site after adsorption. The interaction between the O radical and the surface releases the largest amount of energy (−4.56 eV after adsorption), followed by HO_2_, OH, H, CH_3_ radicals, and O_2_ with the adsorption energy of −3.79, −3.61, −3.07, −2.21, and −1.09 eV, respectively. The energy barriers have been listed in the left corner of Figure 6. The H-abstraction of the chemisorbed C_2_H_2_ by OH and O_2_ have similar barriers, which are 1.39 and 1.43 eV, while the H-abstraction reactions by adsorbed HO_2_, H, CH_3_, and O radicals need to overcome higher energy barriers, which are 1.64, 2.75, 2.15, and 2.07 eV. The H-abstraction reactions by all the adsorbed radicals considered are exothermic except for H radical, indicating that the H-abstraction reactions on the perfect Cu_2_O (111) surface model is thermodynamically favorable.

The energy profile of the interaction between the chemisorbed C_2_H_2_ species and the pre-adsorbed oxygen and radicals, including H, OH, O, HO_2_, and CH_3_ on the defective Cu_2_O (111) surface model are compared in Figure 7. In general, the energy release of the radicals on the defective surface is lower than those on the perfect Cu_2_O (111) surface model, which is due to the absence of the unsaturated surface Cu_CUS_ site. The adsorption of an atomic O on the defective Cu_2_O (111) surface model releases the highest amount of energy (−3.22 eV), which is located at the bridge site of two surface Cu_CSS_ sites. The H-abstraction of the chemisorbed C_2_H_2_ by the adsorbed O radical need to get over the 0.52 eV energy barrier, which is the lowest among all the considered radicals and O_2_, and it is also lower than that on the perfect Cu_2_O (111) surface model. The bridge site of two neighboring Cu_CSS_ sites is also the adsorption site for OH radicals, and the energy barrier of the H-abstraction process by OH is 0.91 eV. The energy barriers of the H-abstraction by other adsorbed radicals, including O_2_, HO_2_, and H radicals are 0.97, 0.98, and 1.17 eV, respectively, which are more active than the same processes on the perfect Cu_2_O (111) surface model. Therefore, the Cu_CUS_ defect could improve the Cu_2_O (111) surface activity toward the H-abstraction of C_2_H_2_ via the LH mechanism.

### 3.6. Temperature Dependence of Elementary Reaction Rate Constants

The above-mentioned discussions are based on the DFT calculation results at 0 K, and the rate constants at elevated temperatures are more relevant to the real circumstances, and the temperate dependence of elementary reaction rates is discussed in this section. The Gibbs free energy of activation (Δ*G*) of the elementary reactions, including the C_2_H_2_ chemisorption, C_2_H_2_ direct decomposition, and H-abstraction of C_2_H_2_ by O_2_, are shown in Figure 8. Δ*G* denotes the Gibbs energy difference between the transition state and the initial state, and a larger Δ*G* corresponds to a smaller reaction rate constant at a given temperature according to the transition state theory. Except for the reaction of H-abstraction by O_2_ on the defective Cu_2_O (111) surface model, the Δ*G* of all the considered reactions are positively correlated to the temperature. For the reactions of C_2_H_2_ chemisorption and the direct decomposition of the chemisorbed C_2_H_2_ into an adsorbed C_2_H and an atomic H species, the perfect Cu_2_O (111) surface model shows better performance over the defective Cu_2_O (111) surface model, while the H-abstraction process is more favorable on the defective Cu_2_O (111) surface model than the perfect one, so the defective surface could facilitate the H-abstraction process of C_2_H_2_ by O_2_.

The reaction rate constants of elementary reaction steps are further calculated based on transition state theory, and the rate constants including C_2_H_2_ chemisorption, C_2_H_2_ direct decomposition, and the H-abstraction of C_2_H_2_ by O_2_ are presented in Figure 9. The C_2_H_2_ chemisorption process is more active on the perfect Cu_2_O (111) surface model than on the defective Cu_2_O (111) surface model, and the reaction rate is about one order of magnitude higher. The C_2_H_2_ direct decomposition rate constants are the slowest regardless of the perfect or the defective Cu_2_O (111) surface models compared with other elementary reaction steps considered in Figure 9, and the rate on the defective surface model is slower than on the perfect surface. The rate constant of H-abstraction by O_2_ on the defective surface model is remarkably higher than on the perfect one, and the rate constant is faster by about 4.4 orders of magnitudes at 1000 K. Therefore, the perfect Cu_2_O (111) surface model is more favorable for the C_2_H_2_ chemisorption process, while the defective surface model could be beneficial for the H-abstraction reaction of C_2_H_2_ by O_2_ over the considered temperature range from room temperature to 1000 K. In terms of the reaction rate constants, the direct decomposition of C_2_H_2_ is less likely to proceed compared with the H-abstraction process. Therefore, the chemisorption and the succeeding H-abstraction process by O_2_ could be the possible partial oxidation reaction pathway of C_2_H_2_ on the Cu_2_O (111) surface model. The chemisorption of C_2_H_2_ is the rate-determining step on the defective Cu_2_O (111) surface model, and the H-abstraction by the O_2_ process is the rate-determining step on the perfect Cu_2_O (111) surface model.

The calculated rate constants are then converted into the Arrhenius form with the *A*, *n*, and *E* parameters provided in Table 1 for future heterogeneous kinetic modeling works.

## 4. Conclusions

The adsorption and oxidation of C_2_H_2_ on the perfect stoichiometric and Cu_CUS_-defective Cu_2_O (111) surface models are studied using DFT calculations. The perfect Cu_2_O (111) surface model is active in adsorbing the C_2_H_2_ molecule with 1.13 eV adsorption energy and an energy barrier of 0.70 eV to form the chemisorption, while the adsorption of C_2_H_2_ on the defective Cu_2_O (111) surface model only releases 0.20 eV. The energy barrier of the C_2_H_2_ chemisorption on the defective surface model is 0.81 eV which is close to that of the perfect model. The reaction rate of H-abstraction of the chemisorbed C_2_H_2_ on the defective surface model is much faster than on the perfect surface model with the energy barrier decreasing from 1.43 to 0.97 eV at 0 K. The H-abstraction of C_2_H_2_ by O_2_ is about 4.4 orders of magnitudes faster at 1000 K on the defective Cu_2_O (111) surface model than on the defective Cu_2_O (111) model. Therefore, the surface defect featuring the absence of surface unsaturated Cu_CUS_ sites significantly facilitates the H-abstraction of C_2_H_2_ by O_2_ process. The partial oxidation of C_2_H_2_ on the Cu_2_O (111) surface model is likely to proceed by the chemisorption of C_2_H_2_ and a succeeding H-abstraction process by O_2_. C_2_H_2_ chemisorption is the rate-determining step on the defective Cu_2_O (111) surface model and the H-abstraction process is the rate-determining step on the perfect Cu_2_O (111) surface model. The activity of H-abstractions of C_2_H_2_ via the LH mechanism by various radicals follows the order of OH > O_2_ > HO_2_ > O > CH_3_ > H from high to low on the perfect Cu_2_O (111) surface model. On the defective Cu_2_O (111) surface model, the activity follows the order: O > OH > O_2_ > HO_2_ > H > CH_3_.

## Figures and Tables

**Figure 1 molecules-27-06748-f001:**
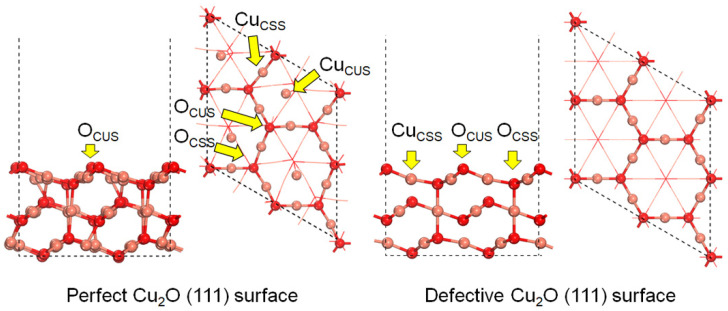
Structures of the perfect and the defective Cu_2_O (111) surface models.

**Figure 2 molecules-27-06748-f002:**
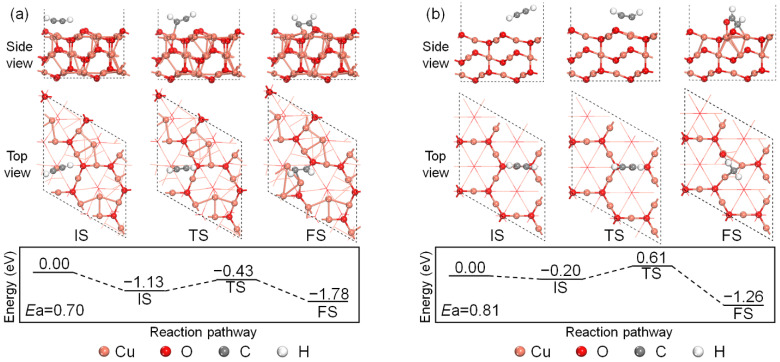
Energy profile of C_2_H_2_ chemisorption on the (**a**) perfect Cu_2_O (111) and (**b**) defective surface models.

**Figure 3 molecules-27-06748-f003:**
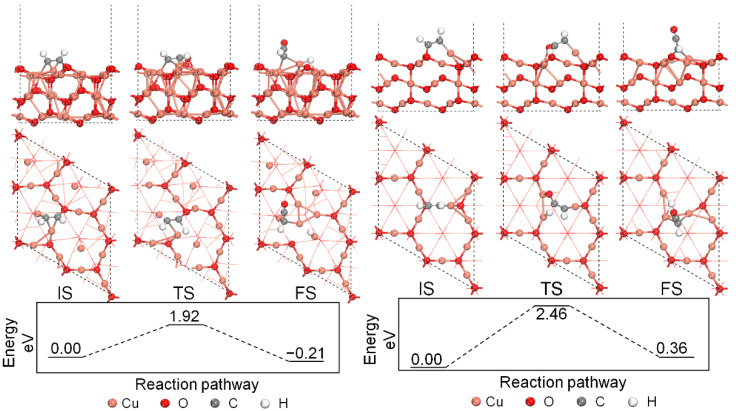
Energy profile of C_2_H_2_ direct decomposition on the Cu_2_O (111) surface models.

**Figure 4 molecules-27-06748-f004:**
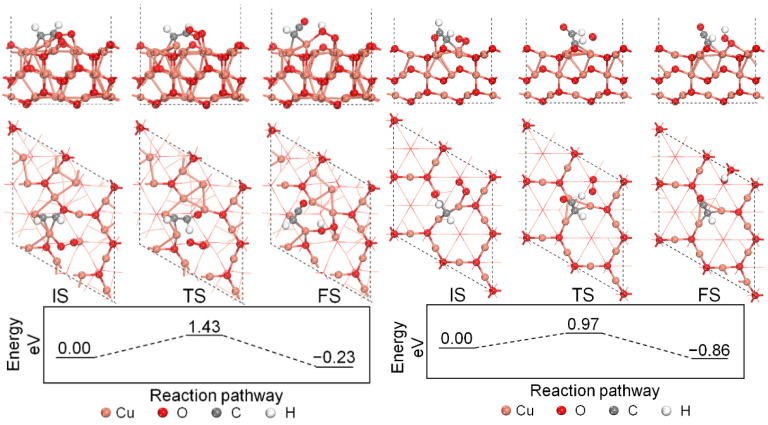
Energy profile of C_2_H_2_ H-abstraction by O_2_ on the perfect and defective Cu_2_O (111) surface models.

**Figure 5 molecules-27-06748-f005:**
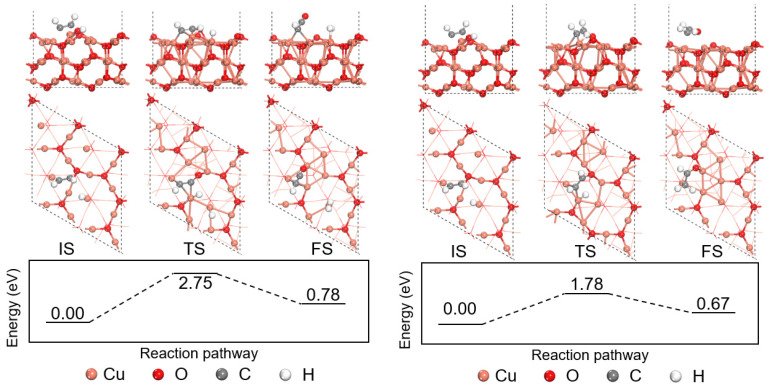
Energy profile of chemisorbed C_2_H_2_ reaction with atomic H into CH_2_CHO and HCCO on the perfect Cu_2_O (111) surface model.

**Figure 6 molecules-27-06748-f006:**
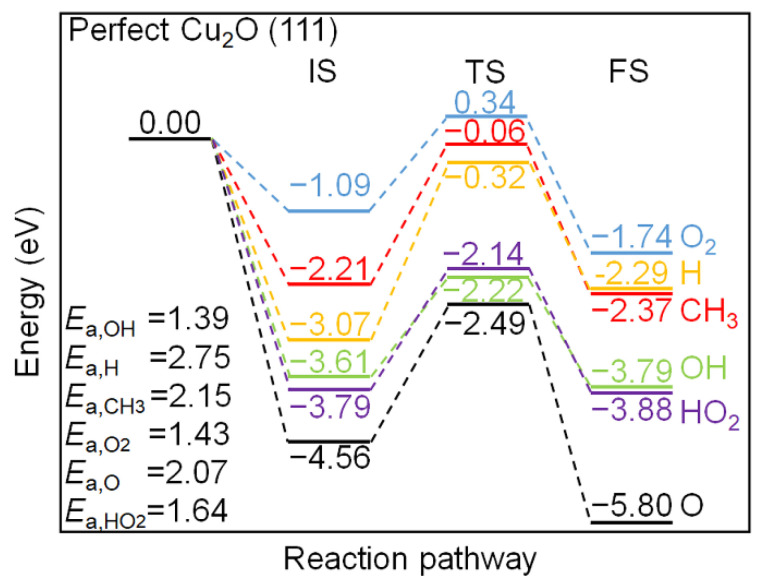
Energy profiles of C_2_H_2_ H-abstraction by various radicals on the perfect Cu_2_O (111) surface model.

**Figure 7 molecules-27-06748-f007:**
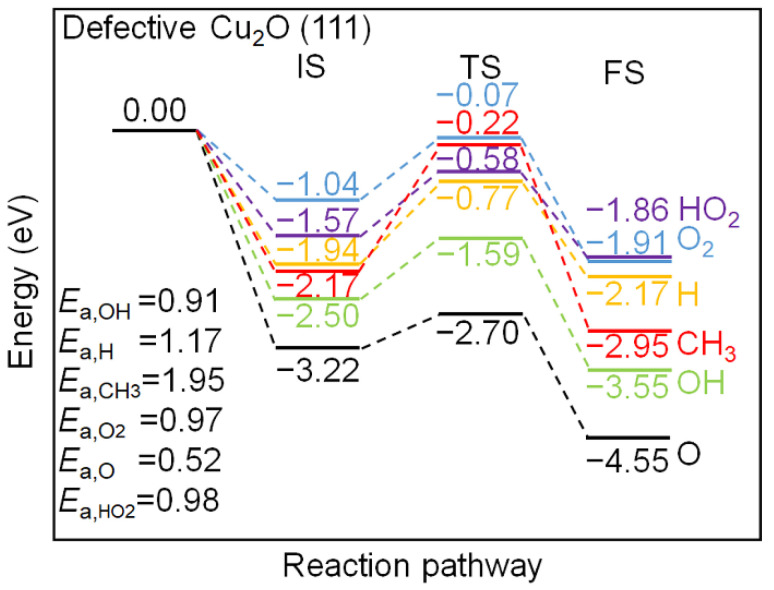
Energy profiles of C_2_H_2_ H-abstraction by various radicals on the defective Cu_2_O (111) surface model.

**Figure 8 molecules-27-06748-f008:**
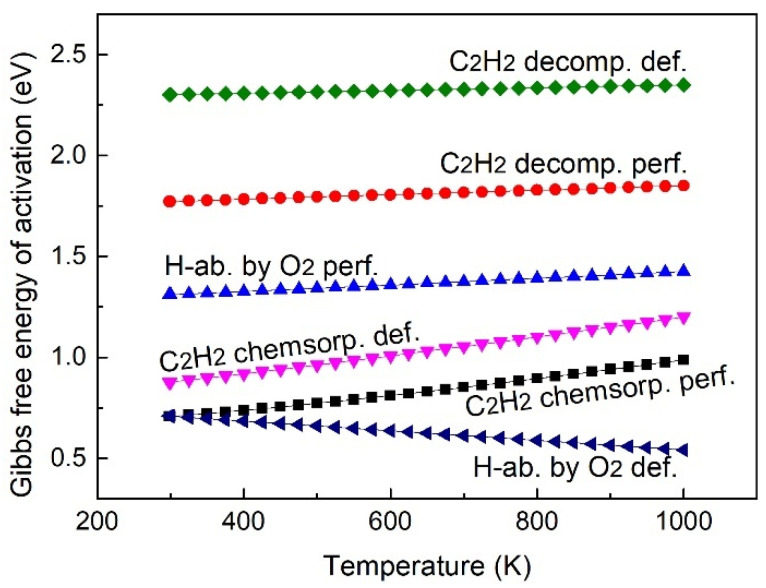
Gibbs free energy of activation of elementary reactions.

**Figure 9 molecules-27-06748-f009:**
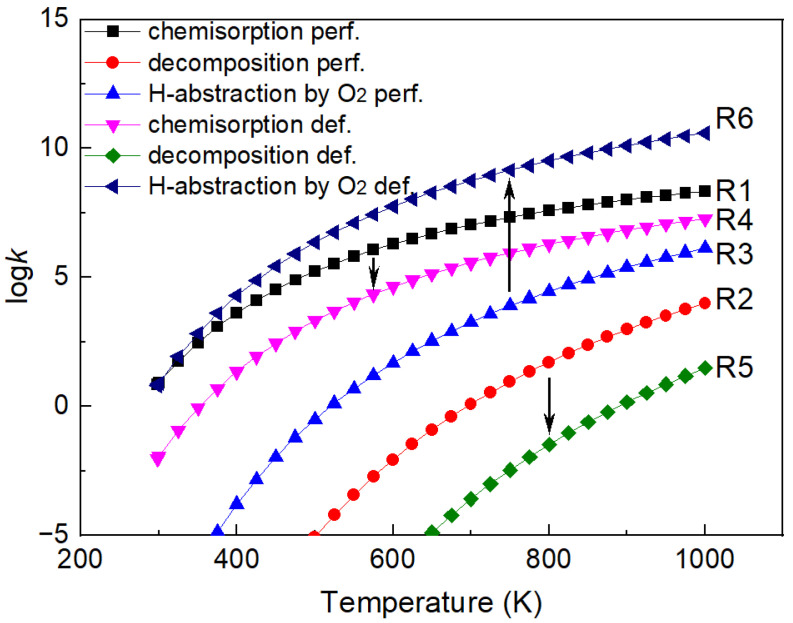
Reaction rate constants of C_2_H_2_ chemisorption, decomposition, and H-abstraction processes.

**Table 1 molecules-27-06748-t001:** Calculated rate constants of C_2_H_2_ elementary reactions on the perfect and defective Cu_2_O (111) surface models, units are in K, kcal, mol, s, and cm.

No.	Elementary Reactions	*A*	*n*	*E*
R1	C_2_H_2_ chemisorption on the perfect Cu_2_O (111) surface	5.11 × 10^13^	−0.687	15.117
R2	C_2_H_2_ decomposition on the perfect Cu_2_O (111) surface	1.15 × 10^10^	0.906	40.193
R3	C_2_H_2_ H-abstraction on the perfect Cu_2_O (111) surface by O_2_	3.16 × 10^10^	0.689	29.443
R4	C_2_H_2_ chemisorption on the defective Cu_2_O (111) surface	1.98 × 10^11^	−0.021	18.130
R5	C_2_H_2_ decomposition on the defective Cu_2_O (111) surface	5.13 × 10^7^	1.656	51.245
R6	C_2_H_2_ H-abstraction on the defective Cu_2_O (111) surface by O_2_	8.15 × 10^12^	0.588	18.679

## Data Availability

The data presented in this study are available in this article.

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
