# Peer review of "A Density Functional Theory and Microkinetic Study of Acetylene Partial Oxidation on the Perfect and Defective Cu2O (111) Surface Models"

_molecules, 2022, doi:10.3390/molecules27196748_

Round 1
Reviewer 1 Report
Your manuscript is generally well written and presented. I have a few comments and a general doubt about the relevance of your work though, and thus ask you to carefully address them. Thus, I am recommending a major revision for your work. This doesn’t mean the work needs reformulating, maybe it’s just to improve the “story telling”, because at the moment it doesn’t seem solid enough.
Comment 1
Introduction, first paragraph: incomplete. Reference to acetylene as raw-material of industrial interest should also be mentioned, as well as the extensive work done with heterogenous catalysis in such processes. This should be framed together with the content already present, for the reader to get a more accurate picture.
Comment 2
Introduction (line 60). This sentence is simply wrong. Please reformulate or remove. Furthermore, do the studies you refer have any kind of empirical data? If they are all modelling and/or simulation, then consider reformulate what you state here. In line 62 you use “correlation”?! to address the knowledge needs on surface properties. No, not correlation. You need a good hypothesis with predictive capabilities (aka theory).
Comment 3
Did I understand correctly that the simulation results up to fig. 7 were obtained at 0 K? I really do not understand the point of this… Could you please develop further?
Comment 4
Line 279. You wrote: “…a larger ΔG corresponds to a smaller reaction rate constant at a given temperature…” Can one really assess reaction rates from Gibbs free energy?!
Comment 5
I have 2 fundamental doubts about the relevance of this work: 1) about half of the data presented was obtained at 0K, and I don’t see any relevance for it, and 2) the lack of experimental validation of, at least, the basic assumptions behind the models studied. A few experiments and measurements could confirm or deny the need for much of the calculations you did. Please develop further on these two comments, as at the moment I am not sure about the relevance of your work, but admit I may be misunderstanding what you are trying to convey.
Author Response
Reviewer 1
Your manuscript is generally well written and presented. I have a few comments and a general doubt about the relevance of your work though, and thus ask you to carefully address them. Thus, I am recommending a major revision for your work. This doesn’t mean the work needs reformulating, maybe it’s just to improve the “story telling”, because at the moment it doesn’t seem solid enough.
Comment 1
Introduction, first paragraph: incomplete. Reference to acetylene as raw-material of industrial interest should also be mentioned, as well as the extensive work done with heterogenous catalysis in such processes. This should be framed together with the content already present, for the reader to get a more accurate picture.
Response: The authors appreciate the reviewer for the professional comments on the manuscript. Reference to acetylene as a raw material of industrial interest has been discussed and mentioned in the revised manuscript.
Added descriptions: Acetylene (C2H2) is a kind of important industrial raw material used for various purposes including oxyacetylene welding, cutting, illuminant, soldering metals, signaling, precipitating metals, particularly copper, manufacture of acetaldehyde, acetic acid, etc [1].
Comment 2
Introduction (line 60). This sentence is simply wrong. Please reformulate or remove. Furthermore, do the studies you refer have any kind of empirical data? If they are all modelling and/or simulation, then consider reformulate what you state here. In line 62 you use “correlation”?! to address the knowledge needs on surface properties. No, not correlation. You need a good hypothesis with predictive capabilities (aka theory).
Response: The sentence mentioned by the reviewer has been removed (line 60 in the introduction part). In this work, the results were obtained based on DFT calculations, which do not depend on empirical data. We have rephrased the sentence in line 62 from “The development of high-performance catalysts requires an insightful understanding of the correlation between surface morphology and its activity toward the surface reaction processes of interest.” to “Good command of the surface oxidation mechanism could be beneficial for the development of high-performance catalysts” We hope the revised statement could address the reviewer’s concern.
Comment 3
Did I understand correctly that the simulation results up to fig. 7 were obtained at 0 K? I really do not understand the point of this… Could you please develop further?
Response: Thank you for your comment. The figures up to Fig. 7 were calculated at 0 K. The electronic properties, initial state, transition state, and final state structures serve as the basis to obtian the stable structures, and electronic properties, and to calculate the temperature effect on the respective reaction routes. This concern is similar to the first question in Comment 5, and we have explained it in more detail together in the response to comment 5.
Comment 4
Line 279. You wrote: “…a larger ΔG corresponds to a smaller reaction rate constant at a given temperature…” Can one really assess reaction rates from Gibbs free energy?!
Response: Thanks for your comment. According to the transition state theory, the elementary reaction rate constants were calculated according to , where k is the rate constant, kB is the Boltzmann constant, h is the Planck constant, T is the temperature, is the Gibbs free energy difference between the initial state and the transition state. Therefore, according to the rate constant equation, the rate constant is inversely proportional to at a given temperature.
Comment 5
I have 2 fundamental doubts about the relevance of this work: 1) about half of the data presented was obtained at 0K, and I don’t see any relevance for it, and 2) the lack of experimental validation of, at least, the basic assumptions behind the models studied. A few experiments and measurements could confirm or deny the need for much of the calculations you did. Please develop further on these two comments, as at the moment I am not sure about the relevance of your work, but admit I may be misunderstanding what you are trying to convey.
Response: Thank you for the comment.
- Regardingthe comment on “half of the data was obtained at 0 K and the relevance”.
Usually, for a DFT study, the structures and the target systems to study were optimized using the selected methods and the basis set at the selected accuracy according to the Schrödinger equation. Therefore, for the DFT study, the structures and electronic properties of the studied system were first explored, which provide the basic and general information of the target system. The energy and the transition state structures were calculated and searched based on the optimized initial states and the final states at 0 K. That is the reason why a part of the results was firstly presented at 0 K, which provides information to the readers about the stable structures and electronic properties of the reaction process. In addition, to further study the temperature influence on the reaction The vibrational frequencies were calculated, and the thermal corrections to the energy were included in this way. Therefore, after introducing the results at 0 K, the results at elevated temperatures were presented.
- Regardingthe comment on “the lack of experimental validation”.
This work is a theoretical attempt to investigate the reaction mechanism of C2H2 on the Cu2O (111) catalyst surface using DFT calculations. The selected calculation methods and basis set are well accepted in surface reaction studies, which could provide some new insights into the surface reaction mechanism. The lack of experimental results is mainly because it is hard to prepare a catalyst sample with a single surface plane without any defect. Indeed, if such experiments could be performed, it would be very helpful in validating the calculated results, and it would be our next research endeavor in solving this problem.
Reviewer 2 Report
Comments:
In this manuscript, the authors studied the adsorption and oxidation mechanism of C2H2 on the Cu2O(111) surface model and the mechanism of Cu2O catalytic removal of C2H2 through density functional theory. The phenomenon of C2H2 chemisorption by a series of free radicals, for instance H, OH, HO2, CH3, O and O2 were compared. This work is of great significance for waste gas purification and industrial safety, because it is beneficial to understand the heterogeneous reaction mechanism of C2H2 and optimize the treatment of C2H2 in industrial production. I recommend this work to be published in Molecules. Before its publication, some minor issues should be addressed as follows:
1, the Ocss site is not shown in Figure 1. The authors should indicate Ocss in Figure 1.
2, In Figure 6, the release sequence of IS free radicals interacting with the surface IS not consistent with the figure. It is very hard to follow the description in Line 240 to line 249. For example, -3.54 eV refer to HO2, but in Fig. 6, HO2 shows -3.79 eV. This part should be carefully revised.
Round 2
Reviewer 1 Report
Thank you for your work in addressing my comments. I am satisfied with your responses and thus am recommending the publication of your manuscript in its present form.